# MoME: Mixture of Multimodal Experts for Generalist Multimodal Large Language Models

**Leyang Shen**[*]  **Gongwei Chen**[*]  **Rui Shao**[†]  **Weili Guan**  **Liqiang Nie**[†]

School of Computer Science and Technology, Harbin Institute of Technology, Shenzhen

{chengongwei, shaorui, guanweili, nieliqiang}@hit.edu.cn

## Abstract

Multimodal large language models (MLLMs) have demonstrated impressive capabilities across various vision-language tasks. However, a generalist MLLM typically underperforms compared with a specialist MLLM on most VL tasks, which can be attributed to task interference. In this paper, we propose a mixture of multimodal experts (MoME) to mitigate task interference and obtain a generalist MLLM. Our MoME is composed of two key components, a mixture of vision experts (MoVE) and a mixture of language experts (MoLE). MoVE can adaptively modulate the features transformed from various vision encoders, and has a strong compatibility in transformation architecture. MoLE incorporates sparsely gated experts into LLMs to achieve painless improvements with roughly unchanged inference costs. In response to task interference, our MoME specializes in both vision and language modality to adapt to task discrepancies. Extensive experiments show that MoME significantly improves the performance of generalist MLLMs across various VL tasks.

## 1 Introduction

Recently, Multimodal Large Language Models (MLLMs) [39, 35, 48, 31, 67] have witnessed remarkable progress. With the help of Large Language Models (LLMs) [13, 63, 1] and Modality Encoders [50, 30, 17, 52, 70], MLLMs demonstrate excellent multimodal comprehensive abilities, especially in solving a wide range of vision-language (VL) tasks [3, 42, 37, 55, 57, 56], such as Image Cpation, Visual Question Answering, Referring Expression Comprehension, and Optical Character Recognition. However, it is increasingly acknowledged that a generalist MLLM tends to have lower performance compared to a specialist MLLM on most VL tasks [12, 8], as depicted in Fig. 1 (a). This phenomenon can be attributed to task interference, a fundamental and crucial issue in multi-task learning [16, 41, 64].

There are some preliminary attempts [15, 6, 12, 5, 8, 58] to address this issue in instruction-tuning MLLMs. The most promising direction [12, 5, 8, 58] among these attempts is to use a mixture of experts (MoE) in MLLMs, aiming for each expert to specialize in several tasks. However, these works only investigate MoE in LLMs and primarily concentrate on textual differences between tasks, overlooking the equally important visual information. As shown in Fig. 1 (b-c), we analyze the distribution of various VL tasks across both vision and language modalities. It is evident that images from different groups of VL tasks exhibit distinct feature distributions, as do texts. Inspired by this observation, we argue that handling task interference needs to comprehensively exploit task differences in both vision and language modalities.

---

[*]Equal contribution

[†]Corresponding authors

38th Conference on Neural Information Processing Systems (NeurIPS 2024).

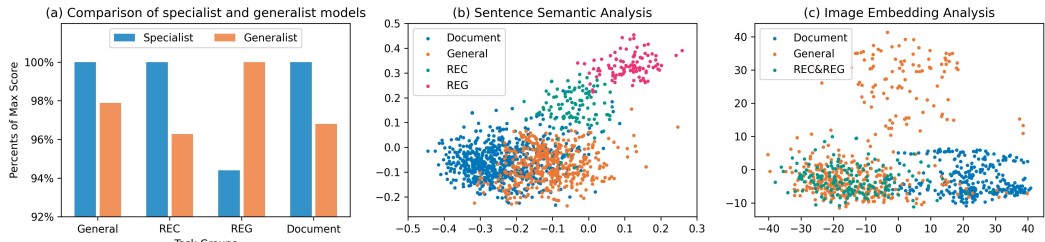

Figure 1: **VL data distribution visualization and model performance comparisons.** Experimental results in (a) show that a generalist model trained on a mixed dataset underperforms most specialist models trained on separate task groups. The feature distributions visualized in (b) and (c) show significant discrepancies across VL tasks in both images and instructions.

To mitigate task interference, we devise a Mixture of Multimodal Experts (MoME) and integrate it into MLLMs. Our MoME consists of a Mixture of Vision Experts (MoVE) for adaptively aggregating features from various vision encoders [52, 17, 50], and a Mixture of Language Experts (MoLE) for leveraging multiple sparsely-activated parameter-efficient adapters. To avoid feature mismatch in different vision encoders, we propose an adaptive deformable transformation (ADT) module in MoVE and use it to transfer features of vision encoders into a unified-length sequence of feature vectors. Our ADT module combines adaptive average pooling and deformable attention [71] to obtain compressed and self-enhanced visual features. After feature transformation, our MoVE uses an instance-level soft router to modulate and aggregate transformed visual features according to the instructions. Our MoLE introduces several parameter-efficient adapters [9] as experts and integrates them by using an instance-level sparsely-activated router. Due to the utilization of adapters, MoLE can be integrated into each feed-forward network layer of an LLM and only incurs a few computational costs with consistent performance gains.

To comprehensively evaluate the multimodal understanding ability of MoME, we collect an amount of VL tasks to form an instruction-tuning dataset and split them into four groups. Extensive experiments show that both MoVE and MoLE can consistently improve performance across all groups of tasks. Notably, MoVE can achieve an average performance gain of 12.87 points across all VL tasks, and improve by over 20 points on the "Document" group. Furthermore, we visualize the expert load distributions of MoVE and MoLE across various tasks. The experts in both MoVE and MoLE exhibit a relatively clear specialization in different groups of VL tasks. For example, the "Document" group of tasks has a strong preference for the "Pix2Struct" vision expert. The expert specialization is strong evidence to demonstrate that our MoME dynamically selects experts to adapt to task differences and mitigate task interference.

Our main contributions are summarized as follows:

- We propose MoME by simultaneously designing mixtures of experts tailored for both the vision encoder and the LLM, resulting in generalist MLLMs with the ability to combat task interference.

- Through statistical analysis, we demonstrate that our MoME specializes in both vision and language modality, effectively adapting to the varying requirements of different VL tasks.

- Extensive experiments show that our MoME possesses excellent multimodal understanding abilities and achieves superior performances across various groups of VL tasks.

## 2 Related Work

### 2.1 Vision Encoders in MLLMs

Vision encoders play important roles in the perception ability of recent MLLMs by encoding visual information into visual tokens, enabling LLMs to understand information on visual modalities. Most Multimodal Large Language Models (MLLM) use CLIP-ViT [52] as their vision encoder to provide the basic image-level perception of an image for LLMs, which is useful for tasks such as image caption and general VQA. However, Tong et al. [62] have found that CLIP-ViT struggles to encode some visual patterns, severely limiting perception and preventing MLLMs from becoming generalist.

To alleviate this, recent works [33, 62, 27] integrated different vision encoders [52, 17, 50, 70, 29] into a single MLLM, which enhanced the perception of MLLM. However, none have effectively mitigated the interference among different visual features, resulting in sub-optimal utilization of each encoder. Differently, we explore the task differences and propose MoVE to perform self-enhanced transformation and adaptive routing among features from different encoders, achieving consistently high performances across vast VL tasks.

## 2.2 Mixture of Experts in Large Models

Mixture of Experts (MoE) [24] is a type of structure with multiple expert networks working together, where each expert is responsible for a part of the knowledge space. By only activating specific experts adaptively during inference, MoE can reduce resource consumption and enhance reasoning speed, which is useful for LLM. Existing MoE LLM [18, 26, 14] usually replace the feed-forward network (FFN) with the MoE layer. Each MoE layer consists of a router and multiple expert networks and each token is assigned to several expert networks by the router. MoE LLMs tend to outperform dense models with the same inference activate parameters.

In addition to its effectiveness in foundation models, some works [12, 8, 58, 20, 32] have utilized MoE in the visual instruction tuning [36] phrase of MLLMs to mitigate task interference, aiming for each expert to specialize in several tasks. Some of them replicate the original FFN in LLMs, while others insert multiple low-rank adaptation [22] modules in parallel with the original FFN, converting LLM into multi-expert architecture. However, they primarily concentrate on LLM but overlook task interference within the visual perceiving process of MLLMs. In contrast, we comprehensively exploit task interference in both vision and language modalities and propose MoME to mitigate them with specialized vision and language experts.

# 3 Methods

To design a generalist MLLM with powerful multimodal understanding capabilities, we begin by analyzing task interference, a common challenge in current MLLMs (Section 3.1). Then, we propose our Mixture of Multimodal Experts and introduce its main components in Section 3.2.

## 3.1 Analysis of Task Interference

Task interference is a fundamental and crucial issue in multi-task learning. As current generalist MLLMs are trained with a number of tasks, they naturally suffer from this issue especially when the number of tasks increases. To investigate this issue in a scenario of MLLMs, we will analyze the feature distribution and the performance change of MLLMs on a tailored instruction-tuning dataset.

To demonstrate the external manifestations of task interference, we first construct a mixed instruction-tuning dataset with various VL tasks and split all tasks into four groups. The performance comparisons between MLLMs trained on the mixed dataset and each task group are illustrated in Fig. 1 (a). It is shown that a generalist model trained on the mixed dataset underperforms specialist models on three task groups. We conclude that the generalist model suffers from a notable task interference problem.

In the era of Large models, there are some attempts to handle task interference from various perspectives, such as instruction, architecture, and dataset configuration. Here, we focus on the mainstream direction, architecture design, and try to explore a robust architecture to combat task interference. In terms of architecture, most existing works prefer a mixture of experts in LLMs. The experts can be feed-forward networks or parameter-efficient modules. However, we argue that this paradigm of architectural design is sub-optimal as visual and textual information should be given equal importance.

In Fig. 1 (b-c), we investigate the feature distribution of various task groups on vision and language modalities, respectively. All samples of each task group are fed into vision and text encoders to produce modality features. These features are transformed by using PCA and then visualized to show the distribution. We observe that the feature distributions of different task groups exhibit significant differences in both the vision and language modalities. As mentioned above, the textual differences can be addressed by multiple experts in LLMs, but visual differences lack effective handling. In the following section, we will introduce our Mixture of Multimodal Experts, which simultaneously handles visual and textural differences between tasks to mitigate task interference.

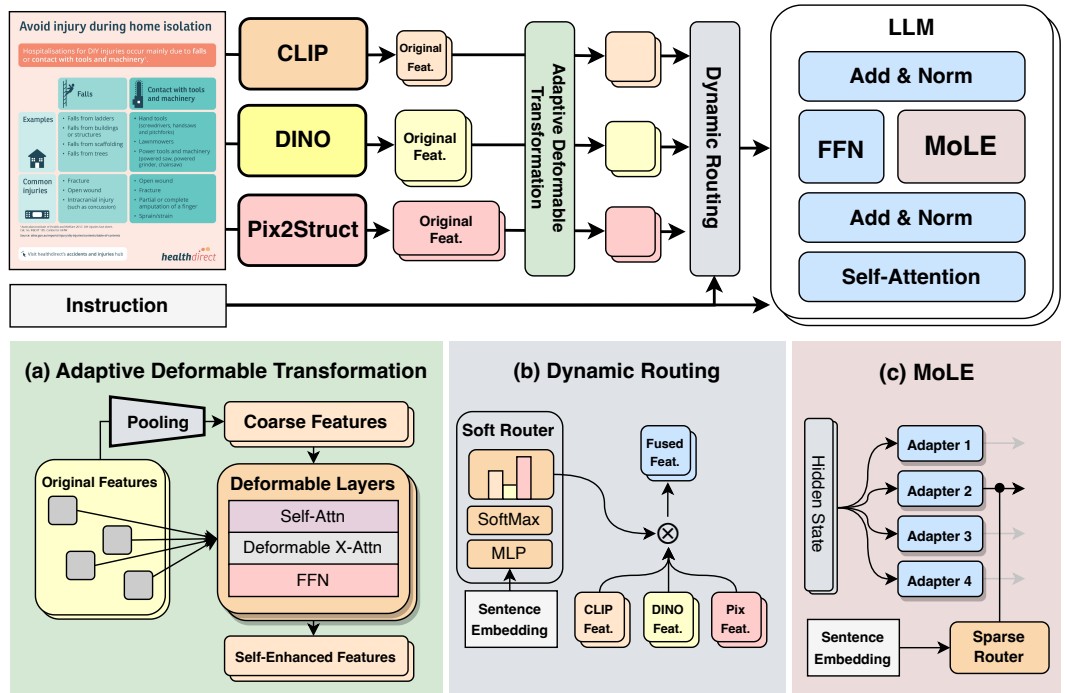

Figure 2: **The overall architecture of the proposed MoME.** The model obtains compressed and self-enhanced visual features from distinct vision encoders through adaptive deformable transformation (a) and aggregates them by dynamic routing (b). The MoLE blocks (c) are integrated into each FFN layer of LLM to improve multitasking capability with little cost.

## 3.2 Architecture

As illustrated in Figure 2, we present our novel MoME architecture that dynamically mixes vision and language experts. The proposed architecture aims to adaptively aggregate visual information (3.2.1) and select LLM pathways (3.2.2) based on the given instructions.

### 3.2.1 Mixture of Vision Experts

Before introducing our MoVE architecture, we will present a pilot study that inspires us to design MoVE. Specifically, we design three MLLMs (consists of vision encoder, projection, and LLM), which share the same architecture except vision encoders. These three MLLMs use three distinct vision encoders, CLIP, DINOv2, Pix2Struct, respectively. After training them using the same data and strategies, we found that MLLMs with different vision encoders excelled in specific tasks, as presented in Fig. 3. the MLLM with CLIP-ViT is good at general tasks and regional caption tasks. the MLLM with DINOv2 excels in REC, a visual grounding task. the MLLM with Pix2Struct is outstanding in text-intensive document understanding tasks. However, each model had weaknesses and none could achieve uniformly excellent performance across all tasks.

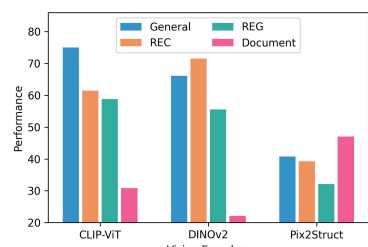

Figure 3: Comparison of MLLMs with different vision encoders.

Inspired by the above study, we propose MoVE to combine various off-the-shell vision encoders and effectively align and aggregate their visual features. The key components in MoVE are an adaptive deformable transformation module and an instruction-based soft router. The former aims to align the features from various vision encoders, and the latter seeks to modulate and aggregate transformed features based on the instructions.

**Adaptive Deformable Transformation.** Given the diversity in architecture and training methods of different vision encoders, the issue of mismatched visual representations in terms of sequence length and feature space becomes significant. While current researches [62, 27, 33] often focus on models like CLIP-ViT [52] and DINO [50], which share similar data processing pipeline, the mismatch problems are less important and frequently overlooked. However, the aspect ratios of Pix2Struct [30] feature shapes vary depending on the input image. Simply combining them through padding and addition will lead to the misalignment among visual tokens and the damage of visual information. To tackle this challenge, we innovatively design an adaptive deformable transformation (ADT) module that effectively transforms features from diverse vision encoders $f$ into a unified-length feature sequence $\hat{f}$, ensuring more coherent and informative visual representations for subsequent aggregations.

As illustrated in Fig. 2 (a), the ADT module consists of a 2D adaptive pooling layer and $M$ deformable attention layers ($\mathbb{D}$). It attempts to automatically select the corresponding information from original features $f$ to refine the coarse features obtained by 2D adaptive pooling $\mathcal{D}(f)$,

$$\hat{f} = \mathbb{D}^M(\mathcal{D}(f), f) \tag{1}$$

Inspired by Deformable DERT [71], we choose deformable cross-attention for its 2D sampling mechanism, which is ideal for interactions among visual feature maps of varying shapes. Meanwhile, it converges fast and has computational and memory efficiency. Specifically, each deformable layer consists of a self-attention layer, a deformable cross-attention, and a feed-forward layer. The crucial select operation occurs in the deformable cross-attention layer, which takes the output of the upper self-attention as query $q \in \mathbb{R}^{L \times C}$, samples the original feature map $f \in \mathbb{R}^{H \times W \times C}$, and outputs result $\mathcal{O} \in \mathbb{R}^{L \times C}$. In this module, the first step is to generate attention weights $w \in \mathbb{R}^{L \times N_h \times N_p}$ and $L$ sets of 2D sampling points, denoted as $p$, from the input queries $q$. For each set, there are $N_h \times N_p$ points, where $N_h$ signifies the number of attention heads and $N_p$ represents the number of points sampled by each head. The sample points and attention weights generation process is as follows,

$$p_{ijk} = (\mathcal{P}_p(q_i)_{jk} + R_i), i \in \{1, \ldots, L\}, j \in \{1, \ldots, N_h\}, k \in \{1, \ldots, N_p\} \tag{2}$$

$$w = \mathcal{P}_w(q) \tag{3}$$

where $\mathcal{P}$ denotes the linear projector and $R \in \mathbb{R}^{L \times 2}$ is a learnable vector called reference point. Then, the corresponding information is sampled by attention heads from the value feature maps $\mathcal{P}(f)_j$ projected and split on the last dimension for each head. The sampling mechanism of each attention head is as follows,

$$o_{ij} = \sum_{k=1}^{N_p} w_{ijk} \cdot \text{Sample}(\mathcal{P}_v(f)_j, p_{ijk}), i \in \{1, \ldots, L\}, j \in \{1, \ldots, N_h\} \tag{4}$$

The results of multiple attention heads are concatenated and projected to obtain the output feature $\mathcal{O}$, which is the input of subsequent feed-forward layer.

$$\mathcal{O} = \mathcal{P}_o(o) \tag{5}$$

**Instance-level Soft Router.** Since images from different groups of VL tasks exhibit distinct feature distributions, there is no one-fits-all strategy to aggregate them. To address this, we propose to generate a customized fusion ratio for each sample based on its instruction.

Specifically, we devise an instance-level soft router $G_s$, as depicted in Fig 2 (b). The router generates corresponding ratios for the visual representations from different vision encoders, followed by a weighted sum of these visual representation features $\hat{f}$, which can be formulated as,

$$G_s(I) = \text{SoftMax}(W_2(\sigma(W_1 I + b_1)) + b_2) \tag{6}$$

$$\mathcal{F} = \sum_{i=1}^{N} G_s(I)_i \times \hat{f}_i \tag{7}$$

where $N$ is the number of vision experts and $\sigma$ denotes GeLU [21]. The router is a multilayer perceptron (MLP) followed by a SoftMax layer to ensure that the sum of the weights equals 1. The instruction is first passed through Sentence-BERT [53] to extract sentence embedding $I$, which is then fed into the router.

Table 1: **Comparison of various MoVE strategies.** "Gen." and "Doc." respectively denote average performances on General and Document. "Avg." means the average of the four group scores. The best performances are marked as bold.

| # | Encoder | Strategy | | Performance | | | | |
|---|---------|----------------|-------------|-------|-------|-------|-------|-------|
|   |         | Transformation | Aggregation | Gen.  | REC   | REG   | Doc.  | Avg.  |
| 1 | CLIP    | AvgPool        | -           | 75.04 | 61.42 | 58.79 | 30.84 | 56.52 |
| 2 | DINO    | AvgPool        | -           | 66.09 | 71.52 | 55.58 | 22.10 | 53.82 |
| 3 | Pix     | AvgPool        | -           | 40.75 | 39.26 | 32.11 | 47.05 | 39.79 |
| 4 | MoVE    | AvgPool        | Addition    | 70.36 | 74.89 | 57.55 | 32.83 | 58.91 |
| 5 | MoVE    | ADT            | Addition    | 74.35 | 76.93 | 61.01 | 39.23 | 62.88 |
| 6 | MoVE    | ADT            | Router      | **79.05** | **81.92** | **63.82** | **52.77** | **69.39** |

### 3.2.2 Mixture of Language Experts

We introduce MoE architecture in LLM, aiming for each expert to specialize in several tasks. However, conventional MoE methods typically incorporate multiple parallel FFN layers in one block, significantly increasing training costs and memory consumption. To meet the multi-task learning needs in the instruction tuning stage, we incorporate several parameter-efficient adapters [9] parallel to FFN. Each adapter enhances the original FFN with task-specific understanding capabilities, thus effectively enhancing the multitasking abilities with a few computation costs.

We insert an MoE block parallel to each FFN layer in LLM. As depicted in Fig. 2 (c), The MoE block consists of several low-rank adapters and an instance-level sparsely-activated router $G_h$. The adapter is designed as a bottleneck structure for computational efficiency, featuring a down-projection layer $\mathcal{P}_{down}$, a ReLU layer $\sigma$, and an up-projection layer $\mathcal{P}_{up}$. Moreover, a learnable scalar $\mathtt{s}$ is multiplied in the output to weigh the importance adaptively. The whole low-rank adaptation process is as follows,

$$y = \mathtt{s} \cdot \mathcal{P}_{up}(\sigma(\mathcal{P}_{down}(x))) \tag{8}$$

The router is an MLP network followed by a top-1 gate function to ensure the output is a one-hot vector $G(I) \in \{0, 1\}^K$. The router generates the selection based on the sentence embeddings $I$ used in MoVE. Each sample is routed to the corresponding adapter to calculate the adapted value $o$, which can be further added to the output of the original FFN. The whole process of the MoLE block is as follows,

$$o = \sum_{k=1}^{K} G_h(I)_k \times y_k \tag{9}$$

where K denotes the number of experts.

## 4 Experimental Results and Analysis

We collect 24 datasets and categorize them into four groups for instruction-tuning and evaluation, the details of which can be found in Appendix B.

### 4.1 Analysis on MoVE

We conduct experiments on MLLMs with different vision encoders under the same training strategy to verify the effectiveness of the two key components in MoVE: ADT and router. Experimental results are summarized in Table 1. We take the multitasking performances of MLLMs with a single vision encoder as our baselines. The adaptive average pooling is applied to the visual representation from DINO and Pix2Struct, ensuring that the lengths of visual tokens fed into LLM are consistent. Experiments #1-3 show that MLLMs using CLIP, DINO, and Pix2Struct as vision encoders exhibit distinct strengths in General, REC, and Document tasks, respectively. Moreover, in the REG task, which requires both captioning and visual grounding abilities, the performance of MLLMs with CLIP and DINO significantly surpasses that of those using Pix2Struct. We can conclude that a single vision encoder cannot meet the visual perception needs of all tasks.

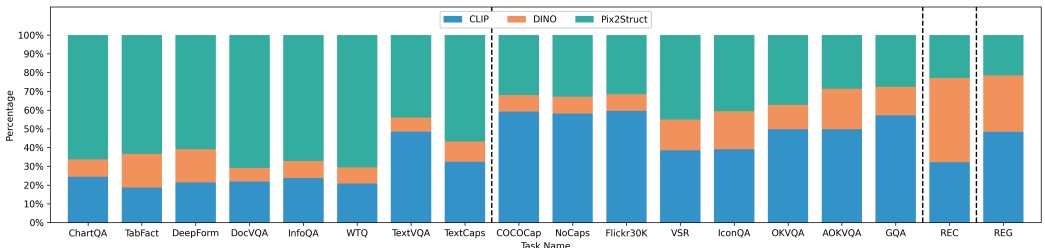

Figure 4: **Distribution of vision experts routing results.** In each bar, the lengths of different colors represent the frequency with which each expert is selected.

To make different vision encoders work together in perception, we first aggregate different visual representations by addition (#4). The average performance does improve compared with models with single vision encoders (#1-3). However, such a straightforward method does not bring much improvement, and even some sub-items have declined. This is due to the mismatch and interference among different visual features, which severely compromise their respective visual information. To make visual features aligned well and reduce information loss, we first replace the pooling with the proposed ADT network. As demonstrated in Experiment #5, ADT consistently enhances performance across four task groups, yielding an average improvement of 4 points. Based on the transformed visual features, we further replace the naive addition mixture process with the router to modulate and aggregate them adaptively according to instructions (#6). This achieves an impressive performance that significantly outperforms the addition method and the methods with a single vision encoder across all sub-tasks. Additional experiments on the deformable mechanism can be found in Appendix C. These experimental results prove that ADT and router are crucial and effective components of MoVE to mitigate interference and optimally utilize the capabilities of each visual expert.

**Visualization of Routing Results.** To provide a deeper understanding of the MoVE's adaptive routing process, we visualize the routing outcomes across various tasks. Since the feature scale varies across different vision encoders, we cannot directly consider the output of the router as expert importance. Instead, we integrate the output features from vision encoders, taking the magnitude of the weighted feature vector as the importance metric. Because the final aggregated result will lean towards the side with the larger magnitude.

As displayed in Fig. 4, for tasks that involve text recognition and graph understanding, such as ChartQA and DocVQA, the features from the Pix2Struct encoder are dominant. In contrast, the model utilizes more CLIP features in image-level VQA tasks like COCOCapion [11], NoCaps [2], and Flickr30K [69]. Notably, for TextCaps [59], a task that requires two kinds of visual information, the router shows a preference for balancing the CLIP and Pix2Struct branches. For tasks that focus on visual grounding ability like REC [42], and REG [37], the model uses more DINO features to perceive region-level visual information. These observations indicate that MoVE can adaptively modulate the features transformed from various vision encoders.

## 4.2 Analysis on MoLE

We conduct ablation experiments to explore the best practice of MoLE, which are summarized in Table 2. We take the model with a single adapter in each FFN (#1) as baseline, which suffers severe task interference. Then, we replace the plain adapter with MoLE. As summarized in Table 2 #2-4, we test three kinds of MoLE routers with different inputs: token hidden states (MoLE-T), sentence embedding (MoLE-I), and a mixture of both (MoLE-IT). The token hidden states and sentence embedding are concatenated on the last dimension as the input for the MoLE-IT router. The experimental results indicate that all MoLE variations outperform the plain adapter, with the sentence-embedding-based router achieving the highest average performance.

We also explore two strategies for expert load balance in MoLE, which are tabulated in Table 2 #5-6. MoLE-I+GS introduces variability to the router by adding Gumbel-distributed noise to the logits [25], aiming to prevent certain experts from never being selected. MoLE-I+LB uses auxiliary loss [18] to

Table 2: **Comparison of different MoLE configurations** "-T", "-I", and "-IT" respectively represent MoLE with routers based on token, instance, and both. "GS" and "LB" represent the implementation of Gumble Softmax [25] and Load Balance [18] based on the MoLE-I, respectively.

| # | VE | LE | Gen. | REC | REG | Doc. | Avg. |
|---|------|-------------|-------|-------|-------|-------|-------|
| 1 | CLIP | Adapter | 74.50 | 63.80 | 59.24 | 31.73 | 57.32 |
| 2 | CLIP | MoLE-IT | 75.61 | 65.63 | 59.17 | 32.46 | 58.22 |
| 3 | CLIP | MoLE-T | 75.35 | 66.09 | 58.09 | 32.12 | 57.91 |
| 4 | CLIP | MoLE-I | 75.62 | 66.95 | 60.90 | 32.27 | 58.94 |
| 5 | CLIP | MoLE-I + GS | 74.87 | 63.97 | 59.00 | 31.69 | 57.38 |
| 6 | CLIP | MoLE-I + LB | 75.42 | 64.74 | 59.48 | 32.05 | 57.92 |
| 7 | MoVE | Adapter | 79.05 | **81.92** | 63.82 | 52.77 | 69.39 |
| 8 | MoVE | MoLE-I | **79.65** | 81.58 | **64.83** | **53.69** | **69.94** |

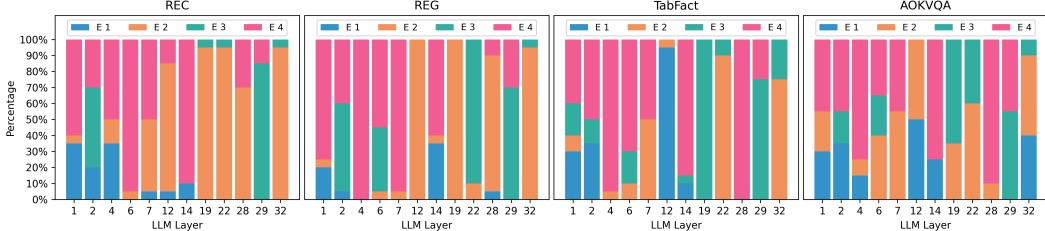

Figure 5: **Distribution of language experts routing results.** The figures depict the expert load conditions of four selected datasets. In each bar, the lengths of different colors represent the frequency with which each expert is selected.

impose load balancing. However, we find that these methods are not suitable for our MoLE as they perform worse than the plain MoLE-I configuration.

Through comprehensive comparative experiments, we choose MoLE-I as the default configuration for MoME. By training based on the MoVE model, MoME further enhances the multitasking capability of MLLM, as shown in Experiments #7 and #8.

**Visualization of routing results.** To provide a deeper understanding of our MoLE module, we sample several instances from each dataset and calculate their average routing outcomes. In Fig. 5, we present the expert load conditions of four selected datasets, each representing a kind of VL task. At the beginning, the router assigns equal probabilities to each expert as it is randomly initialized. However, after training, the routing distributions of MoE blocks vary significantly across tasks, as shown in Fig. 5. This means the language experts in our MoLE module gradually specialize in distinct task domains during training. In the inference process, the router adaptively chooses the optimal expert to achieve strong multitasking capabilities. Meanwhile, we can observe strong resemblances in the expert routing results of similar tasks, which are further analyzed in Appendix D.

### 4.3 Comparison with state-of-the-art MLLMs

We summarize the evaluation results of MoME and other MLLMs with similar resource consumption on popular VL tasks in Table 3. The results show that our MoME method achieves promising outcomes on most datasets compared with other generalist and MoE MLLMs, especially on TextCaps, Flicker30K, and IconQA.

### 4.4 Qualitative Analisys

We present several visualized examples of our MoME model from distinct dataset groups along with their MoVE and MoLE routing results in Fig. 6. In the REC case, DINOv2 accounts for nearly 50% among vision experts, providing fine-grained visual information. So the model can recognize the blue

Table 3: **Comparison with state-of-the-art MLLMs with similar resource consumption.** MoME achieves superior performances on most datasets and is capable of a broader range of VL tasks.

| Model | Doc VQA | Chart QA | Text Caps | Text VQA | Flickr 30K | Icon QA | VSR | GQA | Ref COCO |
|---|---|---|---|---|---|---|---|---|---|
| Shikra-7B [7] | - | - | - | - | - | - | - | - | 80.2 |
| Ferret-7B [68] | - | - | - | - | - | - | - | - | 82.5 |
| IBLIP [15] | - | - | - | 50.7 | 82.4 | 43.1 | 54.3 | 49.2 | - |
| LLaVA-v1.5 [35] | - | - | - | 58.2 | - | - | - | **62.0** | - |
| LION [5] | - | - | 108.8 | - | 87.4 | 54.89 | **73.8** | 51.6 | - |
| DocPedia [19] | 47.1 | 46.9 | - | **60.2** | - | - | - | - | - |
| MoE-LLaVA [32] | - | - | - | 50.2 | - | - | - | 61.1 | - |
| MixLoRA [58] | - | - | - | 40.0 | - | - | 51.2 | - | - |
| MoCLE [20] | - | - | - | 57.1 | 81.9 | 46.3 | 64.7 | 49.3 | - |
| LLaVA-MoLE [8] | 30.0 | 42.4 | - | - | - | - | - | - | - |
| MoME | **50.8** | **57.2** | **130.8** | 53.2 | **94.6** | **61.4** | 61.9 | 59.7 | **83.2** |

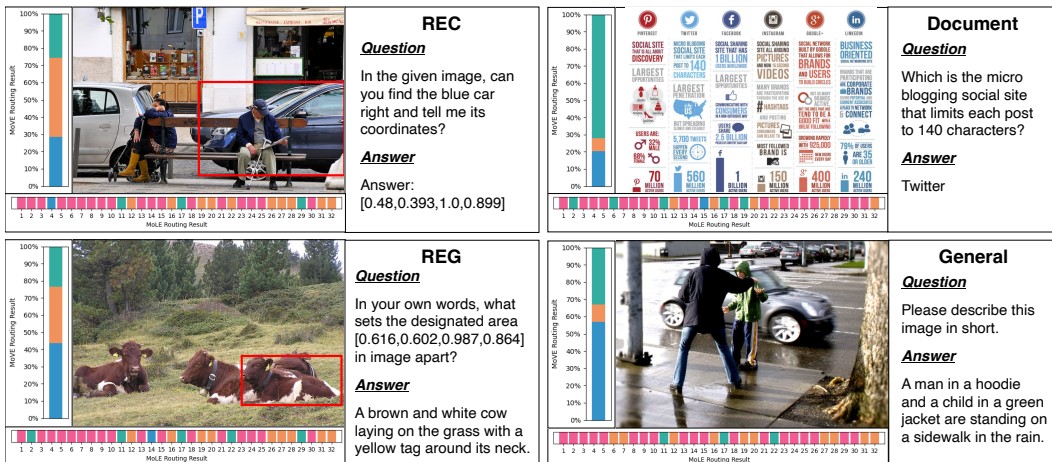

Figure 6: **Visualization of samples along with their routing result distributions.** The MoVE distributions on the left represent Pix2Struct, DINOv2, and CLIP-ViT from top to bottom. MoLE is on the bottom, with different colors indicating different experts.

car in the background and provide its precise bounding box. The Pix2Struct branch accounts for over 70% in the Document case for structured text understanding. The REG case utilizes information from both the CLIP-ViT and DINOv2 to locate objects and generate captions. In contrast, the conventional caption task in the General group only requires an image-level perception, so the CLIP-ViT is dominant. Remarkably, we can observe significant differences among the MoLE routing results. These examples show how MoME selects vision and language experts to adapt to various tasks.

## 5   Conclusion

This work investigates task interference when training a generalist MLLM across various VL tasks. To mitigate it, we propose MoME, which specializes in both vision and language modality to adapt to task differences. Extensive experiments validate the efficiency of MoME as a generalist MLLM.

However, due to the limitations of computing resources, We have not yet expanded our approach to more data and more modalities for experiments. Nonetheless, we believe that the proposed MoME is versatile and can be applied to constructing generalist models in a wider range of multimodal domains. We hope MoME will inspire new research in general-purpose multimodal AI and its applications.

# 6 Acknowledgement

This study is supported by National Natural Science Foundation of China (Grant No. 62306090, No. 62476071, No. 62236003), Natural Science Foundation of Guangdong Province of China (Grant No. 2024A1515010147), and Shenzhen College Stability Support Plan (Grant No. GXWD20220817144428005).

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

# A  Implementation Details

## A.1  Architecture Details

The MoME includes three off-the-shelf vision encoders: CLIP-ViT, DINOv2, and Pix2Struct. For CLIP-VIT, we use ViT-G/14 from EVA [17] without the last layer. The DINOv2 [50] is the official pre-trained version of ViT-L/14 without registers. For Pix2Struct, we use the vision branch of pre-trained Pix2Struct-base model [30]. The ADT consists of a 2D adaptive average pooling layer and a six-layer deformable attention network. The hidden size of ADT is the same as its corresponding vision encoder. We use 8 attention heads and each head samples 4 points in the deformable cross-attention process. In MoLE, the hidden dimension of each adapter is set to 64. We use Vicuna-v1.5(7B) [13] as our pre-trained LLM.

## A.2  Training Details

Our training process comprises two stages. In stage 1, we train the model with MoVE and a single adapter in LLM for 30k steps with a batch size of 64. The learning rate is warmed up linearly from 0 to 5e-4 across 1000 steps and then reduces to a minimum of 0 using cosine decay. The AdamW [38] optimizer is employed with $\beta_1 = 0.9$, $\beta_2 = 0.999$, and a weight decay of 0.05. In stage 2, we load the checkpoint of stage 1 and replicate the weights of adapters to initialize MoLE, while keeping everything else unchanged.

We use a single node with A800 80GB × 8 for all experiments, the entire training is done in one day including stage 1 and stage 2.

# B  Details of Multitasking Benchmark

We collected 24 datasets and categorized them into four groups for instruction-tuning and evaluation, as shown in Fig. 7. For most vision-language (VL) tasks, we used the datasets in both the training and evaluation phases. However, we only use NoCaps for evaluation because it only has an evaluation set, and we exclude the VSR training data due to its simplicity. During the training process, we mix the datasets within the same group into one large dataset, so the probability of each sub-dataset being sampled equals their size as a proportion of the total. However, we ensure the sample ratio of each group dataset is the same. For evaluation, we compute the overall score for each category by averaging its subitem evaluation results. We follow InstructBLIP [15], Shikra [7], and UReader [66] for our evaluation metrics, which are tabulated in Table 5. Notably, the model only takes images as visual information without introducing OCR tokens like InstructBLIP.

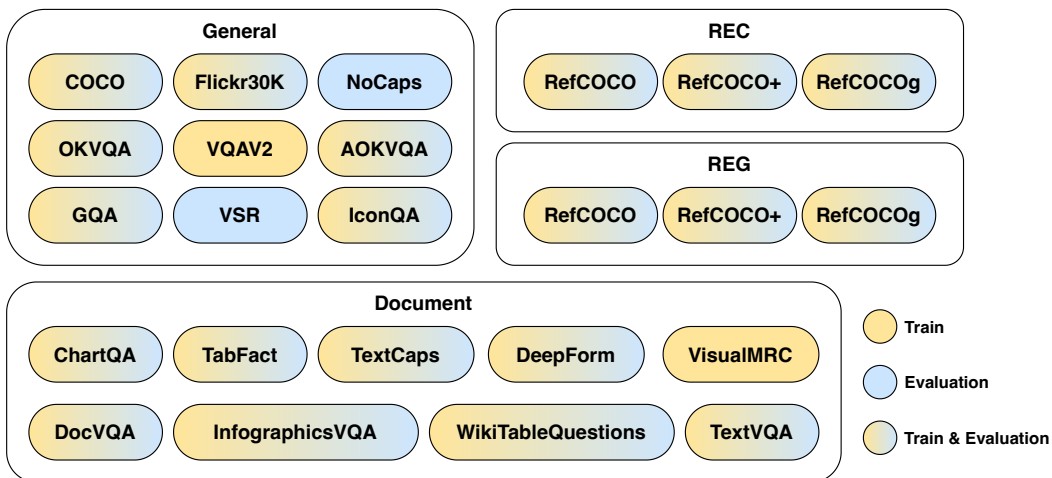

Figure 7: Multitask learning and evaluation datasets and their corresponding categories.

Table 4: **Ablation studies of deformable mechanism.** "Gen." and "Doc." respectively denote average performances on General and Document. "Avg." means the average of the four group scores. The best performances are marked as bold.

| # | Deformable Mechanism | Aggregation | Gen. | REC | REG | Doc. | Avg. |
|---|---|---|---|---|---|---|---|
| 1 | × | Router | 75.27 | 78.17 | 61.64 | 42.46 | 64.39 |
| 2 | ✓ | Router | **79.05** | **81.92** | **63.82** | **52.77** | **69.39** |

## C   Additional Ablation Experiments

To evaluate the effectiveness of the deformable mechanism of ADT, we conduct an ablation experiment by replacing the deformable cross-attention in MoME with standard cross-attention. As shown in Table 4, the model without deformable mechanism (#1) presents much worse performances than the original MoME (#2) consistently. We attribute this decline to the strong inductive bias of deformable attention in processing 2D feature maps.

## D   Additional Visualization and Analysis

We present the routing distributions across different datasets of all MoLE routers in Fig. 8. In general, the routing results differ significantly among different task groups, while the routing preferences are similar within the same task group. From the perspective of router preferences, the datasets can be classified as text-rich (ChartQA - TextCaps), caption (COCOCap - Flickr30K), VQA (IconQA - GQA), REC, and REG. It can prove that the MoLE captures the modularity of the tasks and mitigates task interferences through differential expert routing.

## E   Societal Impacts

MoME utilizes pre-trained large language models (LLMs), which inherently carry limitations from LLMs. These limitations include the potential for generating inaccurate information or biased outputs. To address these issues, we enhance the model's visual perception ability with MoVA and conduct vision-language instruction tuning on high-quality datasets. Despite these improvements, we advise caution and recommend thorough safety and fairness assessments before deploying MoME models in any downstream applications.

Table 5: Summary of the evaluation datasets.

| Task | Dataset | Split | Metric |
|------|---------|-------|--------|
| General | COCOCap [11] | karpathy-test | CIDEr [65](↑) |
| | Flickr30K [69] | karpathy-test | CIDEr [65](↑) |
| | NoCaps [2] | val | CIDEr [65](↑) |
| | OKVQA [44] | val | Accuracy(↑) |
| | AOKVQA [54] | val | Accuracy(↑) |
| | GQA [23] | test-dev | Accuracy(↑) |
| | Visual Spatial Reasoning (VSR) [34] | val | Accuracy(↑) |
| | IconQA [40] | test | Accuracy(↑) |
| REC | RefCOCO [28] | val & testA & testB | Accuracy(↑) |
| | RefCOCO+ [28] | val & testA & testB | Accuracy(↑) |
| | RefCOCOg [43] | val & test | Accuracy(↑) |
| REG | RefCOCO [28] | val & testA & testB | CIDEr [65](↑) |
| | RefCOCO+ [28] | val & testA & testB | CIDEr [65](↑) |
| | RefCOCOg [43] | val & test | CIDEr [65](↑) |
| Document | ChartQA [45] | test | Relax Accuracy [49](↑) |
| | TabFact [10] | test | Accuracy(↑) |
| | DeepForm [61] | test | F1 Score(↑) |
| | DocVQA [47] | test | ANLS [4](↑) |
| | InfographicsVQA [46] | test | ANLS [4](↑) |
| | WikiTableQuestions (WTQ) [51] | test | Accuracy(↑) |
| | TextCaps [59] | val | CIDEr [65](↑) |
| | TextVQA [60] | val | Accuracy(↑) |

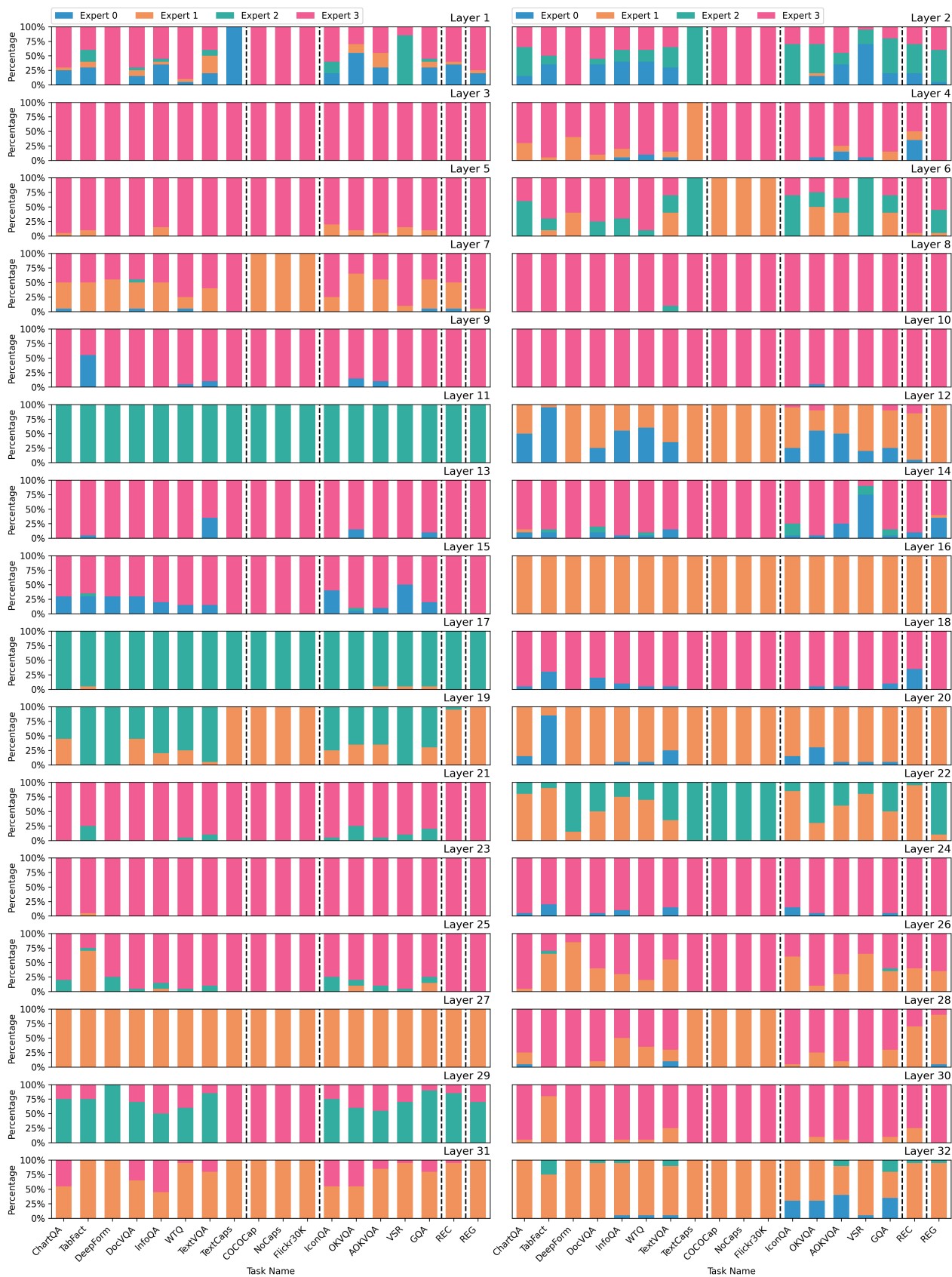

Figure 8: Distribution of all language experts routing results across all tasks.

