# OpenReview forum: "MoME: Mixture of Multimodal Experts for Generalist Multimodal Large Language Models"
_NeurIPS.cc/2024/Conference — NeurIPS 2024 poster_

### Official Review · Reviewer_ZwrT · 2024-06-16

**Soundness:** 2
**Presentation:** 3
**Contribution:** 2
**Rating:** 5
**Confidence:** 4

**Summary:**

The paper proposes to mitigate task interference during multimodal instruction tuning with a mixture of experts, in both the language and the encoder side. The paper is well written, contains insightful analysis and shows improvements over baselines.

**Strengths:**

- Mixture of experts is an important topic that is heavily studied in LLMs but little in multimodal models.
- The problem of task interference is an important problem that is more present in a multimodal setting.
- The paper is well written and easy to follow.
- The proposed approach is well motivated with insightful analysis and show improvements over baselines

**Weaknesses:**

1. While the paper shows improvements over different baselines in Tab.3, the scores are still lagging behind, compared to methods without MoE, such as LLaVA (which the paper builds on top of). Knowing that the proposed approach have significantly more parameters, pretrained for more steps (stage 1) and use different visual encoders.
2. The paper claims that “This means the language experts in our MoLE module gradually specialize in distinct task domains during training.” However the visualization does not support that. For instance, in Fig.5  E4 is used in most tasks types, while E1 is relatively less used. But it is not clear if we can map different experts to different task groups.
3. The paper is not evaluated on common and recent multimodal benchmarks: SEED, MME, MM-Vet, POPE, VQAv2 … that are considered in other methods like LLaVA.
4. The design of MoE are very different for the LLM and visual encoders. Each visual encoder is considered as an expet. Did the authors experiment with typical MoE (replicating FFN) in a single visual encoder?
5. The papers used deformable cross-attention, but I did not find any experiment to support this design choice, compared to using simple cross-attention. Did the authors conduct this experiment?
6. The paper states that load balancing did not help. Any insights on why this is the case? knowing that these losses are typically used in most MoE papers. Also, did the authors encounter any instabilities during training? It is important to discuss these in the paper, as instability is a major problem in MoEs training.
7. The paper main novelty seems in applying MoE also in the vision encoders. This question is also explored in previous works with slightly different context such as [1] and [2]. I found the novelty limited in this regard.

[1] Mustafa, Basil, et al. "Multimodal contrastive learning with limoe: the language-image mixture of experts." Advances in Neural Information Processing Systems 35 (2022): 9564-9576.

[2] Shen, Sheng, et al. "Scaling Vision-Language Models with Sparse Mixture of Experts." Findings of the Association for Computational Linguistics: EMNLP 2023. 2023.

**Questions:**

Please the weaknesses section (e.g. 4-5-6)

**Limitations:**

Some limitations and societal impacts are discussed in the paper or the appendix.

---

> ### Author Rebuttal · Authors · 2024-08-07
>
> **Q1:** Lag behind LLaVA with more parameters, vision encoders, and training time.
>
> **A1:** The original LLaVA includes much fewer tasks than our MoME for training and evaluation, we reported only the results on the shared tasks in Table 3 of the manuscript. **Thus, it is improper and inaccurate to conclude that our MoME lags behind LLaVA.**
>
> In the following, we evaluate the original LLaVA model on our Multitasking Benchmark (which contains many types of VL tasks) for a fair and comprehensive comparison. Meanwhile, we retrain both the original LLaVA model and LLaVA with additional DINO and Pix2Struct encoders using the same datasets and settings as MoME. The results are summarized below:
>
> |  | Gen. | REC | REG | Doc. | Avg. |
> | --- | --- | --- | --- | --- | --- |
> | LLaVA-v1.5-7B (Original) | 69.78 | 46.40 | 71.85 | 22.06 | **52.52** |
> | LLaVA-v1.5-7B (Retrained) | 75.04 | 61.42 | 58.79 | 30.84 | **56.52** |
> | LLaVA-v1.5-7B (w/ DINO&PixStruct) | 70.36 | 74.89 | 57.55 | 32.83 | **58.91** |
> | MoME-7B | 79.65 | 81.58 | 64.83 | 53.69 | **69.94** |
>
> The table shows that **MoME (avg. 69.94 points) has significant advantages over the LLaVA model (Original avg. 52.52 points & Retrained avg. 56.52 points) on the multitasking benchmark.** Additionally, simply adding encoders to LLaVA (avg. 58.91 points) results in limited improvement and lags significantly behind MoME with the same vision encoders, training time, and nearly the same number of parameters. The substantial performance gains of MoME are achieved through the proposed ADT and Dynamic Router, which adaptively mitigate interference.
>
> ---
>
> **Q2:** Visualization of MoLE.
>
> **A2:** Since each LLM layer contains an independent MoLE block, there is no relationship between experts in different layers. Therefore, the statement that "E4 is used in most task types, while E1 is relatively less used" might be misunderstood, as E1 has different meanings across layers. Instead, from  Fig. 5 we can see that the utilization of experts varies significantly across different tasks, indicating the specialization of language experts. For example, REC and REG tasks primarily use E2 in layer 12, while TabFact uses E1, and AOKVQA uses both.
>
> Moreover, from Fig.8, we can observe that the routing results differ significantly among different task groups, while the routing preferences are similar within the same task group, which means different experts specialized in different task groups. We have included an excerpt of Fig. 8 in the rebuttal supplement material, from which we can see clear differences among text-rich (ChartQA - TextCaps), caption (COCOCap - Flickr30K), VQA (IconQA - GQA), REC, and REG tasks.
>
> ---
>
> **Q3:** Multimodal benchmark results.
>
> **A3: Our MoME focuses on multitasking ability and has advantages in benchmarks that contain diverse types of tasks.** However, recent multimodal benchmarks (e.g. MMBench, MME) are primarily organized in a VQA style with multiple choice formats, having a rather limited scope of tasks and instruction templates. For example, all the instructions in MMBench are single-choice with fewer than four options and exhibit high similarity, which does not align with forms of human expression.
>
> In contrast, we evaluate MoME on a multitasking benchmark that contains many types of tasks with diverse instructions and more closely resembles human instructions in a real-world environment. As seen in Table of **A1**, MoME shows significant performance improvements over LLaVA original, LLaVA retrained on Multitasking Benchmark, and LLaVA with more visual encoders.
>
> ---
>
> **Q4:** Design of MoVE (typical MoE)
>
> **A4:** The MoVE module is designed to leverage powerful, off-the-shelf pre-trained vision encoders each specialized in a specific domain, which are more suitable for multi-modal large language models that are designed to be general and versatile.
>
> Moreover, fine-tuning vision encoders was proven by [1] to be resource-consuming and often led to bad results on usually small-scale VL instruction datasets. Consequently, we chose to freeze the pre-trained vision encoders and regard them as vision experts.
>
> [1] Wang, Guangzhi, et al. "What Makes for Good Visual Tokenizers for Large Language Models?."
>
> ---
>
> **Q5:** Experiments supporting deformable cross-attention.
>
> **A5:** We supplemented an ablation experiment by replacing the deformable cross-attention in ADT with cross-attention, which can be found in Table 1 of the rebuttal supplement material.
>
> The table shows MoVE with cross-attention (Avg. 64.39 points) presents much worse performances than ADT (Avg. 69.39 points) consistently. We attribute it to the powerful inductive bias of deformable attention in processing 2D feature maps. We will add this experiment in the revised version.
>
> ---
>
> **Q6:** Load balancing and instabilities of MoLE.
>
> **A6:** We did not encounter instabilities in the training process of MoLE so the load balancing loss did not help. We infer that this is because the lightweight design of adapters makes the MoE block stable. The same phenomenon was also recorded in [1].
>
> [1] Chen, Zeren, et al. "Octavius: Mitigating task interference in mllms via moe."
>
> ---
>
> **Q7:** Novelty limited.
>
> **A7:** The motivation, structure, and context of our method all have significant differences compared to previous works. “LIMoE” and “Scaling Vision-Language Models with Sparse Mixture of Experts.” both employ typical MoE into CLIP encoders to scale up and improve performances. In contrast, the experts of MoVE are different vision encoders each specialized in a specific domain. We proposed an effective way to make use of the powerful off-the-shelf pre-trained vision encoders and mitigate the interference among them. Moreover, we comprehensively explored task differences in both vision and language modalities and significantly enhanced the multitasking capability of multimodal large language models. Therefore, it can be argued that MoME is novel and provides many valuable insights.

---

> > ### Comment · Reviewer_ZwrT · 2024-08-09
> >
> > Thanks for the detailed response. The reviewers addressed some raised points and added additional experiments supporting their approach. Based on their feedback I will increase my score.

---

> > > ### Author Response · Authors · 2024-08-14
> > >
> > > We are happy to hear that we've addressed your concerns, and we thank the reviewer again for the feedback!

---

### Official Review · Reviewer_cdtp · 2024-07-12

**Soundness:** 3
**Presentation:** 3
**Contribution:** 1
**Rating:** 4
**Confidence:** 4

**Summary:**

The paper proposed a MoE design for MLLM, it utilize MoE in both visual encoding procedure and LLM decoding procedure. The paper utilized a dynamic routing module to mix visual features from different experts, and adopted a multi-adapter structure to combine the knowledge of differnent language experts. Widely conducted experiments showcase the effectiveness of the proposed method on various downstream tasks.

**Strengths:**

The paper widely conducted quantitative results on several multimodal downstream tasks to prove the effectness of the proposed MoE design. The paper also adds qualitative results to intuitively showcase the performance.

**Weaknesses:**

The design proposed in this paper seems to be not innovative enough. Its key design philosophy has already been brought out in many previous works. As for MoE in vision, the CVPR 2024 paper “Eyes Wide Shut? Exploring the Visual Shortcomings of Multimodal LLMs” combines CLIP and DINOv2 features to improve visual representations, the arxiv 2023 paper “SILC: Improving Vision Language Pretraining with Self-Distillation” combines the learning objectives of CLIP and DINOv2 for better pretraining outcome. Moreover, the references [12, 5, 8, 54] cited in this paper has already investigated MoE in LLMs, as stated in the introduction section.
It is already a consensus that MoE in MLLM, especially combining multimodal pretrained feature (such as CLIP) and pure visual pretrained feature (such as DINOv2), can improve model performance. Combining a third feature to improve performance (in this paper, the feature from Pix2Struct) besides CLIP and DINOv2 is a pure engineering design, which could not add up to the technical contribution of this paper. Moreover, the ‘Dynamic Router’ proposed in this paper is also a simple MLP network which is widely used for combining different feature. To sum up, it is hard to claim all these designs as a contribution of this paper.

**Questions:**

1.	What is the differnece between this work and the previous works discussing MoE in vision (the several papers I listed in the weaknesses section), how can this paper brough new knowledge or designs that is perviously unknown or not sufficiently expored by the community?
2.	How does the MoE structure affects the inference speed, are there any quantitative results?

**Limitations:**

The authors already adequaately addressed the potential negative societal impact.

---

> ### Author Rebuttal · Authors · 2024-08-07
>
> **Q1:** Key design philosophy has already been brought out in many previous works.
>
> **A1:** The motivation and capabilities of MoVE are completely different from the previous work.
>
> 1. “**Eyes Wide Shut**” introduced an additional DINO encoder and chose to interleave visual features, which ended up with redundant visual features of twice the length. In contrast, we utilized the MoE technique to dynamically aggregate features and keep the token length unchanged, which is more efficient and adaptive.
> 2. “**SILC**” combined the learning objectives of CLIP and DINOv2 by self-distillation to improve image-text contrastive learning. It did not focus on multitasking nor did it use the MoE technique. In contrast, we concentrated on multitasking MLLMs, innovatively utilizing the MoE technique to leverage pre-trained vision encoders and effectively mitigate interference among them.
>
> Therefore, we believe that our MoVE is a new and novel design that is distinct from previous works.
>
> ---
>
> **Q2:** It is already a consensus that MoE in MLLM, especially combining multimodal pre-trained features (such as CLIP) and pure visual pre-trained features (such as DINOv2), can improve model performance.
>
> **A2:** It is a consensus that combining different visual features can improve model performance. However, combining them in a MoE style has not been explored, which is not trivial. Existing MLLM works have demonstrated combining two or more vision encoders can significantly boost the visual perception ability of MLLM, but they just use simple addition or interleave [1]. In our MoVE, we aim to dynamically and adaptively aggregate these visual features according to task demands. Table 1 in the manuscript shows the significant improvement of our MoVE compared to simple addition.
>
> [1] Tong, Shengbang, et al. "Eyes wide shut? exploring the visual shortcomings of multimodal llms."
>
> ---
>
> **Q3:** Combining Pix2Struct features is a pure engineering design.
>
> **A3:** Introducing Pix2Struct is not an engineering design. In contrast, it reveals several representative problems including aggregating features with different shapes and sizes and encoders focusing on different data domains. As stated in the manuscript, the aspect ratios of Pix2Struct feature shapes vary depending on the input image and it focuses on text-rich images. Experiments in Table 1 have shown that simply pooling and aggregating these diverse features can lead to severe interference. Instead, we proposed MoVE to adaptively transform and aggregate different kinds of features from CLIP (224x224), DINOv2 (448x448), and Pix2Struct (arbitrary shape) and achieved significant improvement. The ability to handle features of arbitrary shape makes MoME more versatile and valuable.
>
> ---
>
> **Q4:** ‘Dynamic Router’ is a simple MLP.
>
> **A4:** The ‘Dynamic Router’ is not just a simple MLP; its significance lies in its working mechanism rather than its internal structure. Its working mechanism allows MoVE to dynamically and adaptively aggregate visual features, greatly reducing interference and improving performance by over 6 points, as shown in Table 1 of the manuscript. Furthermore, existing work [1] has verified that the internal architecture has a negligible effect on performance.
>
> [1] Ye, Qinyuan, Juan Zha, and Xiang Ren. "Eliciting and understanding cross-task skills with task-level mixture-of-experts."
>
> ---
>
> **Q5:** Differences from previous works & New knowledge or designs.
>
> **A5:** Compared to multi-modal large language models that are equipped with more than one vision encoder, MoME revealed the severe conflicts among visual features and proposed an efficient and novel method to tackle the interferences within both the transformation and aggregation process. Compared to works that simply employ typical MoE design into vision encoders, we deeply explore the dilemma of multi-modal large language models and propose a framework that can benefit from a variety of pre-trained models while avoiding interferences among them.
>
> In this work, we highlighted the task interference problem in both the textual and visual information, while previous works only investigated MoE in LLMs and primarily concentrated on textual differences between tasks, overlooking the equally important visual information.
>
> ---
>
> **Q6:** Quantitative results of Inference speed changes.
>
> **A6:** For MoVE, the number of trainable parameters is 112.83M, accounting for 1.36% of the total parameters and increasing inference time by 1.076%.
>
> For MoLE, we use lightweight adapters as our experts, which only increase the parameters by 0.668% and inference time by 5.959%.

---

> > ### Comment · Reviewer_cdtp · 2024-08-13
> >
> > I appreciate the authors' efforts to summarize their contributions. However, I still believe that the design philosophy of this work—mixture of vision expert—is pretty much similar to previous works. A simple adjust or compression in token length won't brought new knowledge to the model. Nevertheless, I appreciate the detailed explanation of the paper's design difference comparing to previous works, as well as the inference speed provided by the authors. Therefore, I would like to raise my score to 4.

---

> > > ### Author Response · Authors · 2024-08-13
> > >
> > > Thanks for your reply. We would like to clarify some concepts again and sincerely hope you will reassess the innovation and contributions of our work.
> > >
> > > 1. The innovative aspect of this paper, the “mixture of vision experts,” differs significantly from previous works [1,2,3]. In the context of the availability of numerous pre-trained models, we have innovatively proposed a dynamic and adaptive approach to fuse vision encoders each specialized in a specific domain. To the best of our knowledge, the adaptive mixture of various pre-trained vision encoders has not been explored in the works of MLLM, as these works only consider addition or concat [4,5]. Prior to the era of MLLM, existing works [1,2,3] focused on applying typical MoE methods (experts are several identical sub-networks) within Vision Encoders and training from scratch, which is a completely different technical solution. **We acknowledge that our MoVE shares a similar design philosophy with previous works at a broad conceptual level (effectively utilizing many visual branches in a vision-language model). However, as mentioned above, the differences in technical details (e.g., the combination framework of vision branches, the feature processing method, and the design purpose) are also significant.**
> > > 2. We would like to argue against the contention that our ADT did not "bring new knowledge to the model." As stated in the manuscript and rebuttal, **one of the major issues in combining various pre-trained vision encoders is the misalignment among visual tokens**. This issue is caused by the differences in the pre-training setting and architecture of these vision encoders. Simply combining them (pooling and addition) will result in significant information loss. Thus, we propose Adaptive Deformable Transformation to mitigate the information loss by adaptively refining the pooled features. As shown in Table 1 of the manuscript and the table below, our ADT is very helpful and achieves **an average gain of almost 4 points**. To conclude, we believe that **our ADT is a promising method to resolve the discrepancies of various pre-trained vision encoders**, instead of “a simple adjust or compression in token length”.
> > >
> > > | Strategy | Gen. | REC | REG | Doc. | Avg. |
> > > | --- | --- | --- | --- | --- | --- |
> > > | Pool + Add | 70.36 | 74.89 | 57.55 | 32.83 | **58.91** |
> > > | ADT + Add | 74.35 | 76.93 | 61.01 | 39.23 | **62.88** |
> > > | MoVE (ADT + Router) | 79.05 | 81.92 | 63.82 | 52.77 | **69.39** |
> > >
> > > We hope that these explanations can address your concerns, and we would greatly appreciate it if you could consider giving us a higher rating.
> > >
> > > ---
> > >
> > > **Reference**
> > >
> > > [1] Naeem, Muhammad Ferjad, et al. "Silc: Improving vision language pretraining with self-distillation."
> > >
> > > [2] Mustafa, Basil, et al. "Multimodal contrastive learning with limoe: the language-image mixture of experts."
> > >
> > > [3] Shen, Sheng, et al. "Scaling vision-language models with sparse mixture of experts."
> > >
> > > [4] Tong, Shengbang, et al. "Eyes wide shut? exploring the visual shortcomings of multimodal llms."
> > >
> > > [5] Jiang, Dongsheng, et al. "From clip to dino: Visual encoders shout in multi-modal large language models."

---

> > > ### Author Response · Authors · 2024-08-14
> > >
> > > As the interactive discussion window is about to close, we sincerely invite the reviewer cdtp to read our follow-up response. We hope that our explanations effectively address your concerns, and we would appreciate it if you could consider revising your rating based on this information.

---

> ### Author Response · Authors · 2024-08-12
> **Reminder to review the rebuttal**
>
> Dear Reviewer cdtp,
>
> Thank Reviewer cdtp again for the valuable comments. We have provided the response to each of the concerns raised in the review, and we are eager to continue the conversation. As the interactive discussion window will close soon, we kindly invite the reviewer to read our response to see if there are any further questions.
>
> Thank you!
>
> Best regards,
>
> Authors

---

### Official Review · Reviewer_1g6K · 2024-07-12

**Soundness:** 3
**Presentation:** 3
**Contribution:** 3
**Rating:** 6
**Confidence:** 3

**Summary:**

In this paper, the authors introduce a mixture of multimodal experts (MoME) to reduce task interference and develop a generalist MLLM. MoME consists of two main components: a mixture of vision experts (MoVE) and a mixture of language experts (MoLE). MoVE can adaptively adjust features transformed from different vision encoders and boasts strong compatibility with various transformation architectures. MoLE integrates sparsely gated experts into LLMs, achieving seamless improvements while keeping inference costs nearly unchanged.

**Strengths:**

1. The analysis of task interference and mixture of vision experts in this paper is clear, highlighting the necessity of a vision mixture of experts
2. The experiments are well-conducted and quite comprehensive
3. The study demonstrates strong performance on most datasets compared with other generalist and MoE MLLMs

**Weaknesses:**

1. From Table 1, we can see that both ADT and Router have achieved notable improvements. Could you explain the internal mechanisms behind this? I am quite curious as to why the improvements are so significant.
2. In Table 3, many values are missing. Could you add some results from more general multimodal benchmarks to make the experiments more comprehensive, such as MME, MMbench, MM-Vet, LLaVA$^W$, Science$^{QA}$, etc.

**Questions:**

One advantage of MoE is its fast inference speed. Could you conduct an experiment to verify the model's inference speed?

**Limitations:**

yes, the author explains the limitations of their study and potential negative societal impact.

---

> ### Author Rebuttal · Authors · 2024-08-07
>
> **Q1:** Reasons for notable improvements in Table 1.
>
> **A1:** When mixing visual representations from different vision experts, they are first transformed into a unified-length sequence of feature vectors and then aggregated, each step of which will severely damage the visual information.
>
> - Transforming visual representations of different sequence lengths into unified lengths using downsample pooling will cause inevitable information loss since it’s rule-based and static.
> - Visual representations are in different feature spaces due to the diversity in data domains and training methods. They will interfere with each other if simply adding them together and cause information loss.
>
> The proposed ADT and router are dynamic and learnable and can effectively mitigate information loss in both steps:
>
> - Transformation: The ADT module uses deformable attention to compensate for the information loss caused by downsample pooling.
> - Aggregation:  The dynamic router can adaptively aggregate features according to task demands, maximizing the retention of visual information appropriate to each task.
>
> Consequently, the proposed MoVE achieved notable improvement compared to the baseline in Table 1.
>
> ---
>
> **Q2:** General multimodal benchmark results.
>
> **A2:** **Our MoME focuses on multitasking ability and has advantages in benchmarks that contain diverse types of tasks.** However, recent multimodal benchmarks (e.g. MMBench, MME) are primarily organized in a VQA style with multiple choice formats, having a rather limited scope of tasks and instruction templates. For example, all the instructions in MMBench are single-choice with fewer than four options and exhibit high similarity, which does not align with forms of human expression.
>
> In contrast, we evaluate MoME on a multitasking benchmark that contains many types of tasks with diverse instructions and more closely resembles human instructions in a real-world environment. In the Table below, we compare MoME with LLaVA v1.5 on the Multitasking Benchmark. The original LLaVA model performs poorly on the Multitasking Benchmark since it is trained with a limited variety of tasks. To ensure a fair comparison, we retrain it using the same data and settings as MoME. While some improvements are observed, its performance remained significantly lower than ours. MoME exhibits a clear advantage in multitasking due to its effectiveness in mitigating conflicts.
>
> |  | Gen. | REC | REG | Doc. | Avg. |
> | --- | --- | --- | --- | --- | --- |
> | LLaVA-v1.5-7B (Original) | 69.78 | 46.40 | 71.85 | 22.06 | **52.52** |
> | LLaVA-v1.5-7B (Retrained) | 75.04 | 61.42 | 58.79 | 30.84 | **56.52** |
> | LLaVA-v1.5-7B (w/ DINO&PixStruct) | 70.36 | 74.89 | 57.55 | 32.83 | **58.91** |
> | MoME-7B | 79.65 | 81.58 | 64.83 | 53.69 | **69.94** |
>
> ---
>
> **Q3:** Inference Speed.
>
> **A3:** The typical MoE in LLMs has a fast inference speed because of its sparse activation mechanism. However, there are other types of MoE that pursue performance improvements by mitigating the task conflict rather than efficiency [1,2]. Inspired by these works, we designed MoVE and MoLE for better performance with only a slight increase in parameters. Specifically, MoVE and MoLE result in just a 1.36% and 0.668% increase in parameters, and a 1.076% and 5.959% inference speed.
>
> [1] Ye, Qinyuan, Juan Zha, and Xiang Ren. "Eliciting and understanding cross-task skills with task-level mixture-of-experts.".
>
> [2] Zadouri, Ted, et al. "Pushing mixture of experts to the limit: Extremely parameter efficient moe for instruction tuning.".

---

> > ### Comment · Reviewer_1g6K · 2024-08-13
> >
> > Thank the authors for addressing most of my concerns, I will keep my positive rating.

---

> > > ### Author Response · Authors · 2024-08-14
> > >
> > > We are happy to hear that we've addressed your concerns, and we thank the reviewer again for the feedback!

---

### Author Rebuttal · Authors · 2024-08-07

We would like to thank all reviewers (R#1g6k, R#cdtp, R#ZwrT) for their time and efforts in providing constructive feedback. We are very encouraged that reviewers found our work effective (R#1g6k, R#cdtp, R#ZwrT), with a clear analysis of task interference (R#1g6k), comprehensive experiments (R#1g6k, R#cdtp), and insightful analysis (R#ZwrT). We have built an official repository for providing well-structured open-source codes (released upon acceptance).

We have responded to all questions and comments in each review. Additionally, supplement material is included in the attached PDF to help clarify related concerns. We hope these responses provide a more comprehensive view of our paper. Please kindly consider increasing your rating if your concerns have been addressed.

---

### Decision · Program_Chairs · 2024-09-25

**Decision:**

Accept (poster)

**Comment:**

This paper studies the problem of building a multimodal generalist by mixture of experts motivated by the observation of task interference. Existing MoE models tend to focus on the LLM side, but this paper investigates the task inference on the vision representation and proposes ADT and dynamic routing techniques to aggregate dis-similar visual representation. The ablations show that the proposed techniques are effective in improving a multimodal generalists across dis-similar task space.

This paper receives divergent reviews with scores 6, 5, 4. All reviewers appreciate the comprehensive experiments which demonstrates the benefits of the proposed approach, and the analysis which are intuitive (cdtp), clear (1g6k) and insightful (ZwrT).  In addition, the importance of understanding multimodal task interference is noted by ZwrT. Considering the reviews and rebuttal, the main objection to acceptance is raised by reviewer cdtp who considers the novelty incremental relative to previous works such as “Eyes Wide Shut? Exploring the Visual Shortcomings of Multimodal LLMs”, "SILC: Improving Vision Language Pretraining with Self-Distillation" and other works using MoEs in LLMs. The author provided a detailed response and the reviewer adjusted the score accordingly, while maintaining that "the design philosophy of this work—mixture of vision expert—is pretty much similar to previous works" and leaning to reject.

Upon careful inspection of the works cited by cdtp and the author rebuttal, I notice that the two cited papers “Eyes Wide Shut? Exploring the Visual Shortcomings of Multimodal LLMs”, "SILC: Improving Vision Language Pretraining with Self-Distillation" are actually studying very different problems than this paper. The first paper focuses on probing the short comings of the existing VLMs and demonstrate effects of MoF (mixture of features) by simply adding or interleaving features on their benchmarks. The second paper proposes to add self-distillation objective to contrastive learning so that the model is capable of both global/local recognition. Thus, the similarity claimed by cdtp is not very convincing in my humble opinion. In addition, as a principle, I do not think the similarity in high-level design philosophy alone should be the ground for rejection. Having inspected the paper myself and other related multimodal MoE works, I do not think this justification of weak reject are well-supported by evidences.

Given that the other two reviews are positive, I'm leaning to recommend acceptance of this paper given the importance of the problem and the comprehensive analysis/experiments noted by all the reviewers. However, due to the lower number of positive reviews, I'd like us to have more discussion on this paper.